# Influence of the Nature of Aminoalcohol on ZnO Films Formed by Sol-Gel Methods

**DOI:** 10.3390/nano13061057

**Published:** 2023-03-15

**Authors:** Anna Vilà, Alberto Gómez-Núñez, Xavier Alcobé, Sergi Palacios, Teo Puig Walz, Concepción López

**Affiliations:** 1Department of Electronic and Biomedical Engineering, Faculty of Physics, University of Barcelona, Martí i Franquès 1-11, 08028 Barcelona, Spain; 2Institute for Nanoscience and Nanotechnology–IN2UB, University of Barcelona, Martí i Franquès 1-11, 08028 Barcelona, Spain; 3Eurecat—Technology Centre of Catalonia, Av. Universitat Autònoma 23, 08290 Cerdanyola del Vallès, Spain; 4Centres Científics i Tecnològics (CCiTUB), University of Barcelona, C/Lluís Solé i Sabaris 1-3, 08028 Barcelona, Spain; 5Wallbox, Carrer del Foc 68, 08038 Barcelona, Spain; 6Fraunhofer Institute for High-Speed Dynamics, Ernst-Mach-Institut EMI, Ernst-Zermelo-Straße 4, 79104 Freiburg im Breisgau, Germany; 7Department of Inorganic and Organic Chemistry (Section of Inorganic Chemistry), Faculty of Chemistry, University of Barcelona, Martí i Franquès 1-11, 08028 Barcelona, Spain

**Keywords:** ZnO films, sol-gel, aminoalcohol, structural characterization, elemental analysis, IR spectroscopy, XRD, SEM, PL, UV-VIS spectroscopy

## Abstract

Here we present comparative studies of: (i) the formation of ZnO thin films via the sol-gel method using zinc acetate dihydrate (ZAD), 2-methoxyethanol (ME) as solvent, and the aminoalcohols (AA): ethanolamine, (*S*)-(+)-2-amino-1-propanol, (*S*)-(+)-2-amino-3-methyl-1-butanol, 2-aminophenol, and aminobenzyl alcohol, and (ii) elemental analyses, infrared spectroscopy, X-ray diffraction, scanning electron microscopy, absorption and emission spectra of films obtained after deposition by drop coating on glass surface, and thermal treatments at 300, 400, 500 and 600 °C. The results obtained provide conclusive evidences of the influence of the AA used (aliphatic vs. aromatic) on the ink stability (prior to deposition), and on the composition, structures, morphologies, and properties of films after calcination, in particular, those due to the different substituents, H, Me, or ^i^Pr, and to the presence or the absence of a –CH_2_ unit. Aliphatic films, more stable and purer than aromatic ones, contained the ZnO wurtzite form for all annealing temperatures, while the cubic sphalerite (zinc-blende) form was also detected after using aromatic AAs. Films having frayed fibers or quartered layers or uniform yarns evolved to “neuron-like” patterns. UV and photoluminescence studies revealed that these AAs also affect the optical band gap, the structural defects, and photo-optical properties of the films.

## 1. Introduction

Zinc oxide (ZnO) is one of the most attractive metal oxides due to its low cost, non-toxicity, chemical and thermal stability, and unique properties [1,2]. ZnO is a semiconductor with wide bandgap (*E_g_* = 3.37 eV), transparent to visible light, exhibits UV-luminescence at room temperature, and its exciton binding energy (~60 meV) is suitable for short-wave photoelectronic uses [1,2,3,4]. Moreover, ZnO doping allows tailoring its electronic and optical properties and may provide multifunctional materials with improved properties or even new ones with potential interest in other areas, for instance, dilute magnetic semiconductors (DMSs) for spintronics [1,2,4,5,6,7,8]. Besides, ZnO-derived nanomaterials, pellets and thin films are gaining relevance in a wider range of areas (i.e., agriculture and biomedicine [9,10,11,12,13,14,15,16,17]).

The development of ZnO-based thin films with different morphologies, interesting optical, electrical, piezoelectrical, and thermal properties among others and even outstanding catalytic and biological activities is a fascinating research area with a huge expansion in the last years. This is mainly due to their utility in electronics, optics, optoelectronics, engineering, and, especially, for the design of new devices (photodetectors, gas sensors and also biosensors, light-emitting diodes, transparent electrodes and transparent thin-film transistors, among others) [1,2,4,6,11,12,13]. It is widely accepted that they are the most promising candidates to achieve low-cost and non-toxic new devices suitable for novel and improved technologies [1,2,6,11,12,13,14,15,16,17,18,19], (e.g., solar energy conversion) [17,18] and, therefore, they are becoming more and more popular.

ZnO-based thin films can be prepared using a variety of procedures [19,20,21,22,23,24,25,26,27,28,29,30,31,32,33,34,35,36,37,38,39,40,41,42], such as pulsed-laser [20], solvothermal/hydrothermal [21,22], electrochemical [23,24] and ultrasonic-mist-vapor deposition methods [25], spray pyrolysis [26], magnetron sputtering [27], microwave-assisted [28] procedures, and also the sol-gel method (SGM) [2,12,29,30,31,32,33,34,35,36,37,38,39,40], which is one of the most attractive and highly preferred, due to its low cost, simplicity, and feasibility. In general, the SGM requires a Zn(II) salt, an organic compound as stabilizer, and a solvent [25,26,27,28,29,30,31,32]. Among the most common substances that are used as stabilizer are aminoalcohols, especially those having the proper orientation of the N- and O-heteroatoms as to bind to the Zn^2+^ ions forming chelates, or acting as bridging ligands in the precursors and, therefore, preventing the precipitation of Zn(OH)_2_ [25,26,27,28,29,30,31,32,43,44,45,46]. The world-wide long-term sol-gel ZnO precursor forms from zinc acetate dihydrate (ZAD) and ethanolamine (Figure 1, **a**) in 2-methoxyethanol (ME) [25,26,27,28,29,30,31,32,38,39,40]. It is well known that its transformation to ZnO is a complex multi-step process that involves in situ formation of alkoxide or alkoxy complexes, the subsequent hydrolysis, polymerization, and thermal treatment to yield the final ZnO [25,26,27,28,29,30,31,32].

Moreover, recent studies point towards some influence of the stabilizer’s molecular flexibility on bond torsion and breaking, and that, therefore, the substituents may have a strong influence on the decomposition processes of the ZAD-stabilizer system [33], and also on the composition and properties of the films obtained after thermal treatment. The final ZnO obtained by using diethanolamine {HN(CH_2_CH_2_OH)_2_, DEA} as stabilizer retained more N than the one obtained by using aminoalcohol **a** {H_2_NCH_2_CH_2_OH, MEA} under identical conditions, even after calcination at high temperatures. Moreover, peaks due to other intermediate and undesirable species, such as Zn(OH)_2_, were revealed by X-ray diffraction studies in the first case [44], thus indicating that the AA has a crucial role in the sol-gel ZnO formation process and also on the properties of the final material.

In previous reports, we have demonstrated that aminoalcohols (AA), such as (*S*)-(+)-2-amino-1-propanol (**b**) and (*S*)-(+)-2-amino-3-methyl-1-butanol (**c**) (both shown in Figure 1 and closely related to (**a**), as well as the phenyl containing derivatives: 2-aminophenol and aminobenzyl alcohol (Figure 1, **d** and **e**, respectively), are also valuable stabilizers in the sol-gel process [33,34,41,42]. However, there is a lack of information about the role and influence of the nature of these five AAs in the complex and multistep transformation from the precursors to the final ZnO films and in their properties. Since **a**–**e** are valuable stabilizers in ZnO sol-gel formation and have different capability to deprotonate—which affects the formation of the alkoxide or alkoxy-Zn(II) complexes—they appeared to be excellent candidates to find out the effects produced by: a) the nature of the AA (aliphatic (**a–c**) or aromatic (**d–e**)), b) the presence of the substituents Me (in **b**) or ^i^Pr (in **c**) or of the -CH_2_- unit (in **e**) on the ZnO formation processes as well as on stability, crystallographic, morphological, optical, and spectroscopical properties of the final ZnO thin films (**1a**–**1e**). In this work, we present a comparative study of the ZnO thin films obtained by SGM, using ZAD, the AAs (**a–e**), and ME, after thermal treatments at different temperatures.

## 2. Materials and Methods

### 2.1. Materials

The AAs (**a**–**e**, see Figure 1) were obtained from of Acrös Organics, ZAD from Panreac, and ME from Aldrich. All the reagents were used as received.

### 2.2. Film Preparation

A 0.5 g amount of ZAD was treated with the equimolar amount of the selected AA (**a–e**) and afterwards 5 mL of the solvent (ME) was added. The mixtures were stirred at room temperature for 30 min and glass substrates were then drop-coated with the obtained inks, and heated to different temperatures (T = 300, 400, 500, and 600 °C) to produce the films **1a**–**1e**.

### 2.3. Characterization Methods

Quantitative determinations of nitrogen, carbon, and hydrogen content in the samples were carried out by elemental organic analyses (EOA) with Thermo EA Flash 2000 equipment working in standard conditions (helium flow of 140 mL/min; combustion furnace at 950 °C; chromatographic column oven at 65 °C). Infrared (IR) spectra were registered at room temperature with a Nicolet 400FTIR instrument using KBr discs.

X-ray diffraction (XRD) data were obtained with a PANalytical X’Pert PRO MPD alpha1 Bragg–Brentano powder diffractometer using a Cu tube operating at 45 kV and 40 mA, a Johansson type Ge (111) primary focalizing monochromator, and a solid-state strip 1D PIXcel^1D^ detector. High-resolution, high statistics, full angular range, Cu Kα_1_ (λ = 1.5406 Å) data were obtained with the following parameters: 2θ/θ scans from 5 to 140 degrees; step size 0.0263 deg.; measuring time per step 200 s (PIXcel^1D^ active length 3.347 deg.); total measuring time per sample 1.2 h. An automatic divergence slit system and a mask enabled a constant irradiated surface (10 × 12 mm^2^) over the analyzed samples. The diffracting volume was also constant regarding the small and finite thickness (bellow 5 µm) of the characterized layers. Full profile analysis was performed, applying Rietveld refinement [47] for the identified crystalline phases. The refinements have been performed through the TOPAS v6 software [48]. The peak width of each phase was modelled with the double-Voigt approach by considering both the Lorentzian contribution of the crystallite size effect and the Gaussian contribution of the microstrain to the peak width [49]. Preferential orientation corrections were applied by spherical harmonics [50]. The background was modelled with a 15th order Chebyschev polynomial. The instrumental contribution to the diffraction profile was calculated with the fundamental parameters approach [51].

Film morphology was observed by scanning electron microscopy (SEM) with a JEOL JSM-7100F and a JEOL JSM-6510 in planar and cross-section views. The UV-VIS spectra were measured with a Lambda-950 of PerkinElmer, with wavelengths between 190 and 1100 nm for which the precision UV-VIS is 0.05–5 nm and precision near-IR is 0.2–20 nm. The photoluminescence (PL) emission spectra of the films in a range between 370 and 850 nm were obtained at room temperature by exciting the samples (with the 325 nm line of a He-Cd laser, using a power density of 8 kWcm^−2^) and collecting the emitted light with the same experimental setup; the PL intensity of all spectra was corrected by the spectral response of the system.

## 3. Results and Discussion

The first difference detected between the different inks arises from the appearance and stability of the samples obtained immediately after preparation, even before deposition. The aliphatic AAs (**a–c**) samples were white and no significant change was observed after several weeks of storage at room temperature. In contrast, the aromatic sample **d** turned black in a few seconds; while the freshly-prepared sample with **e** was initially whitish but it turned dark brown, nearly black, in a few minutes under identical experimental conditions. These findings suggest that at room temperature the stability of the freshly prepared inks increases according to the sequence **d** < **e** < **a**–**c**. The lower stability of the aromatic systems could be attributed to their stronger photo-reactivity.

Comparative studies of the appearance, IR spectra, composition, and X-ray diffractograms of the films **1a**–**1e** obtained at 400, 500, and 600 °C were carried out in order to achieve additional information on the effect produced by the different AAs on the ZnO–film formation process, their stability, and their properties.

### 3.1. Film Characterization

#### 3.1.1. General Appearance and Chemical Analysis

Pictures of films obtained after the thermal treatments are presented in Figure 2, showing that after annealing at 300 °C, samples **1d** and **1e** were clearly darker than **1a–1c**, which exhibited a yellowish or brownish tonality. This finding is similar to that observed for the freshly prepared samples at room temperature. After annealing at 500 °C, films **1a–1c** turned white; while **1d** and **1e** still had black (for **1d**) or deep-brown (for **1e**) regions. This fact suggests that the aliphatic or aromatic nature of the AAs may be important as to modify the ZnO-formation path (as explained in more detail in the following sections). Finally, after annealing at 600 °C, all films showed the typical white color of ZnO, with **1d** exhibiting higher transparency. As visible in Figure 2, after storage of these samples for two years in the dark and at room temperature, no significant deterioration was observed. This is relevant in view of their potential applications.

IR spectra of the films obtained at each annealing temperature (Figure 3) show that the intensity of the bands due to specific organic units or bonds of the aminoalcohols (i.e., C–H, C–C, or C–N, and C–O) decreased gradually as the temperature increased. For T = 400 °C, the IR spectra of **1a–1c** have a very weak band at 478 cm^−1^ that is characteristic of the crystalline ZnO [52]. In Figure 2 and Figure 3 it can be seen that after annealing at 500 °C: (a) films **1a–1c** exhibited the typical color of the ZnO and (b) the intensity of the band at 478 cm^–1^ increased gradually, while those related to the functional groups of the organic units (mainly those between 1200 and 1700 cm^−1^) decreased. Moreover, annealing at 600 °C caused the absorption band related to the ZnO bond to become the most intense one. We also undertook a parallel study with **1d** and **1e** films (Figure 3), and, as expected, the intensity of the bands observed in the range 1600–1400 cm^−1^ (due to the carbon–carbon stretching vibrations of the phenyl ring [53]) decreased from 300 to 400 °C. It should be noted that at these temperatures the appearance of **1d–1e** samples was markedly different from those of **1a–1c** (Figure 2). The IR spectra of **1d** and **1e** films after heating at 500 °C showed the ZnO band, but its relative intensity was smaller than in the IR spectra of **1a–1c** ones at identical temperature. At 600 °C, this absorption became the most intense one in the studied range.

Elemental analyses of the films **1a**–**1e** obtained at each of the temperatures were also carried out and results are presented in Table 1. As expected, as the annealing temperature increased the contents of nitrogen, carbon, and hydrogen (N, C, and H, respectively) decreased gradually. For the aliphatic derivatives (**1a–1c**) at 500 °C the analytical determinations revealed contents of N and H below 1%, and the percentage of C was relatively low: 0.8% (for **1b**), 2.94% (for **1a**), and 3.6% (for **1c**). Notice that for these samples the appearance as well as the IR studies were consistent with the presence of ZnO.

For the aromatic films **1d–1e**, the C, H, and N contents also decrease with increasing temperature, as expected. For each annealing temperature, the amount of organic residues in aromatic films was higher than for the aliphatic analogues **1a–1c** (see Table 1). Upon heating at 600 °C, the percentages of C, H, and N decreased in all cases, but the amount of carbon in the aromatic samples was clearly higher (0.71% for **1d** and 3.49% for **1e**) than in the aliphatic ones (0.24%, 0.27%, and 0.41% for **1a**, **1b,** and **1c**, respectively). Besides, film **1e** retained more nitrogen (0.43%) than **1a–1d** (0.01%).

Since **1e** can be visualized as derived from **1d** by replacement the -OH unit (of **1d**) by the CH_2_OH one (in **1e**), the comparison of the contents of the residual elements indicates that the presence of the pendant –CH_2_OH arm hinders the “elimination of the organic groups”. All these findings suggest that the purity of the final ZnO films (**1a**–**1e**) obtained at 600 °C increase according to the sequence **1e** < **1d** < **1c** < **1b** ≤ **1a**, indicating that under identical experimental conditions the final ZnO films formed from the aromatic AAs and specially from **e** retain more impurities. This could be related to the higher boiling points of **e** and **d** compared with those of **a–c** (in the range 275–283 °C as compared to 170–186.8 °C). The presence of different amounts of residual C (in all cases) and N (in **1e**) may be important as to modify the number and type of defects and vacancies in the structures and, consequently, the properties and potential applications of these films [1,2,15,16,17,19,20,29].

#### 3.1.2. Structural Properties

In order to achieve further information about the formation of **1a**–**1e** and to elucidate whether the nature of the AA (aliphatic vs. aromatic) could modify the path of the ZnO formation, films **1a**–**1e** formed after the thermal treatments were studied by XRD (Appendix A). For comparison purposes, Figure 4 shows spectra of all films. In this section we will describe first the results obtained for the films derived from the aliphatic AAs (**1a–1c**) and afterwards those isolated from the aromatic ones (**1d** and **1c**).

All observed peaks in the diffractograms of films **1a–1c** after annealing at 400, 500, and 600 °C are indexed with the hexagonal, space group P63mc, ZnO wurtzite type form. ZnO hexagonal wurtzite type is the only crystalline observed phase in these samples. Appendix A depict the indexed observed diffractograms in the main angular range for 2θ from 25 to 75 degrees. It is well known that this hexagonal form is more stable than the cubic sphalerite (zinc-blende) one, and also than the rock salt type, which can only be obtained at high pressures [1,2].

For films **1a–1c** and for each one of the temperatures selected in this study the Rietveld refinements enabled the accurate determination of the lattice parameter *a* and *c* and the estimation of the average crystallite size *D* and lattice strain (*ε*) (according to the double-Voigt approach [54]).

Data presented in Table 2 for the aliphatic-based films (**1a–1c**) show that upon heating the lattice parameter *a* increases, while *c* follows the opposite trend and the values obtained at 600 °C, as well as the *c/a* ratio fall within the ranges expected for the hexagonal wurtzite ZnO: 3.2475 Å ≤ *a* ≤ 3.2507 Å; 5.2042 Å ≤ *c* ≤ 5.2075 Å, and 1.593 ≤ *c/a* ≤ 1.603 [1,2]. The estimated crystallite sizes, *D*, at 600 °C (where ZnO conversion is accomplished) are larger than those at 400 °C while the lattice strain, *ε*, follows the opposite trend. These findings are indicative of fewer microdefects in the grains [55,56] and better quality [40] of films **1a–1c** annealed at 600 °C.

When the XRD studies were performed with the films derived from the aromatic AAs **1d** and **1e**, some interesting and differential features were observed. First, the analysis of data obtained after the treatment at 400 °C revealed the presence of two ZnO forms. One of them was the thermodynamically stable wurtzite form (detected in the XRD of **1a–1c** for all temperatures selected). The other one was identified, after indexing the remaining peaks, as the cubic, space group F-43m, sphalerite form (Appendix A). The Rietveld refinements enabled the semi-quantification [57] of the ZnO phases, hexagonal wurtzite and cubic sphalerite. The relative abundance of the cubic phase decreased when the annealing temperature increased. In fact, analyses of XRD data obtained for films **1d** after the heating treatments at highest temperatures (T = 500 or 600 °C) indicated the presence of the wurtzite form exclusively; while for **1e**, the XRD studies revealed the coexistence of both phases and the presence of the cubic form even after the thermal treatment at 500 °C (Table 2 and Appendix A).

For these films **1d** and **1e,** the Rietveld refinements enabled also the cell parameters determination (*a* and *c* for wurtzite and *a* for sphalerite) and the estimation of the average crystallite sizes, *D*. The sphalerite type phase shows lower values of *D* than the wurtzite type phase (Table 2). At 600 °C, where only the wurtzite form remains, *D* is 30 nm for film **1e** (similar to films **1a**–**1c**) whereas for the **1d** film is higher, 46 nm (Table 2).

In order to produce additional information on the evolution of the precursor to the final ZnO film, we prepared and analyzed film **1e** at 300 °C and, in this case, XRD data indicated the presence of both crystalline forms, but now sphalerite was clearly the predominant phase (80.6% vs. 19.4% wurtzite) (Figure 5). These findings suggest that for **1d** and **1e** films, the formation of the final wurtzite ZnO may proceed through sphalerite form.

It is widely accepted that the sphalerite form can be stabilized by growth on cubic substrates [1,2]. Nevertheless, to the best of our knowledge, no evidence for the presence of the cubic form or its coexistence with the hexagonal one (thermodynamically more stable) has been reported so far. XRD studies of films obtained from the aromatic AAs **1d** and specially **1e** reveal that the ZnO-sphalerite form was the major component at low annealing temperatures (i.e., 400 °C); while for the aliphatic-based ones (**1a–1c**), only the hexagonal wurtzite form was detected in all the cases, even for films obtained at the lowest annealing temperature, T = 300 °C.

Moreover, the crystallite size in **1d** after annealing at 600 °C is larger than that of **1e**. These findings indicate that the presence of the –CH_2_– unit between the –OH unit and the phenyl ring (in **1e**) is important as to modify the dimension of the grains and the microdefects therein.

XRD spectra of films annealed at 600 °C do not show any clear texture, with minor variation between samples. Only a slight preferential (00l) orientation is found for aliphatic-derived films, especially **1b**, which is even less visible for aromatic-based samples.

#### 3.1.3. Surface Morphology and Microstructure

In view of the effects produced by the AA nature (aromatic vs. aliphatic) during the annealing process and knowing the importance of the films’ surface morphology in technological applications, SEM studies were also undertaken. The morphology of films **1a**–**1e** annealed at 300, 400, 500, and 600 °C was visualized using both secondary and backscattered electrons, as shown in Figure 6.

Backscattered electrons (Figure 6b) produce small contrast in all the analysed films, indicating continuous layers with chemical uniformity at the microscopic scale for every sample. Only films **1d** exhibit a different behaviour, being cracked even at annealing temperatures as low as 300 °C. Although a very thin film remains adhered to the substrate, producing a very transparent coverage (see Figure 2), the poor stability induced by AA **d** produces films difficult to be controlled, unsuitable for most applications.

Secondary electron imaging (Figure 6a,c) indicated that ZnO precursors **a–c** and **e** promote films with fractal topography depending on the AA and on the thermal treatment. In particular, annealing at low temperature produce wrinkled layers with morphologies that appear as neural nets, with ridges up to 14 μm thick (sample **1a**) or porous structures (sample **1e**). Annealing at 400 °C tends to increase the number of large features at the expense of suppressing the smallest ones, while treatment at 500 °C gives rise to films with more regular patterns and smaller ridges ranging from 0.8 (**1e**) to 6 μm (**1a**). For higher temperature, this trend seems to be inverted, as annealing at 600 °C promotes the formation of neuron-like prominences that remind us of those observed for low temperatures. A completely different behavior can be observed for sample **1d**, which produced cracked films with smooth surfaces at every temperature. Adhesion is lost from the lowest temperatures, and very thin films remain in all cases, allowing excellent transparency (Figure 2).

A closer look allows ascertaining that those films consist on inhomogeneous, dense, agglomerated particles that accumulate preferentially in the ridges. The thickness of the layers differs depending on the region due to these large fractal structures formed by crystallites that group into larger grains, as reported for MEA [44]. The overall thickness of the layers ranges between 1 (continuous bottom layer) and 5 μm (including protrusions). At 600 °C, larger grains are observed, which can be attributed to a coalescence process that change the film topography, producing less uniform layers with thicker protrusions with diameters between 1 µm (**1c**) and 4 µm (**1b**). In contrast, for lower temperatures, the prominences seem formed by dark-coloured amorphous and nanocrystalline agglomerations highly contaminated with organic residues.

The images clearly show the different structures promoted by the aliphatic and aromatic AAs used. At low annealing temperatures, at which samples have a high concentration of impurities, the relief observed for films **1e** includes rather frayed fibres, very different from the more uniform yarns visible for films **1a** to **1c**. This difference might be related to the presence of a noticeable amount of sphalerite phase in films **1e** annealed at temperatures below 600 °C [58], which disappears at this temperature. The different behaviour is still much more evident for films **1d**, where annealing produces quartered layers that at high temperature leave a very thin transparent film. The instability found for films **1d** prevents its use for producing good-quality ZnO films, and, therefore, it will be disregarded in the optical characterizations that follow.

### 3.2. Optical Studies

#### 3.2.1. UV-Visible (UV-VIS) Spectroscopy

Due to the differences observed in the formation processes, morphologies, and compositions of the ZnO thin films, and the increasing interest in these kinds of materials, a comparative study of the optical properties of the films **1a**–**1e** was undertaken. These properties are related to intrinsic characteristics of the films, such as absorption coefficient (*α*), optical conductivity, bandgap energy (*E_g_*), extinction coefficient (*k*), and refractive index (*n*). Films obtained at 300 °C were not included in this study due to the following reasons: (a) in their IR spectra, the typical band due to ZnO at ca. 478 cm^−1^—that is characteristic of the crystalline ZnO [52]—was, when detectable, extremely weak (see Figure 3) and (b) the analytical determinations revealed the presence of large amounts of C, H, and N (especially for the aromatic-based films with a total content around 38% for **1e**, or even higher for **1d** (*ca.* 52%)), thus suggesting that films processed at 300 °C contained a small amount of ZnO and large quantities of organic fragments. In all cases, spectra were recorded in triplicate in order to ensure the repeatability of the results.

Films derived from the aliphatic AAs **1a–1c** show quite similar UV-VIS behavior (Figure 7), reaching high visible-range transparency as temperature increases. Spectra of films obtained at 400 and 500 °C (which, according to the elemental analyses, still retained small quantities of C, H, and N– residues, see Table 1) exhibited a broad absorption with low resolution below 370 nm. In contrast, spectra of films annealed at 600 °C exhibit an absorption band clearly visible in the range of 355–370 nm (Figure 7). Crystalline ZnO films typically show an absorption band at around 359 nm usually attributed to free excitons [59] that shifts toward a lower wavelength due to quantum confinement.

As seen in Figure 7, the UV-VIS spectra of films **1e** do not follow the same trend and the absorption band at around 359 nm is not clearly identifiable even after annealing at 600 °C. As explained above, films **1d** are quartered by annealing and the very thin films that remain produce very small overall absorption. This is why spectra for films **1d** are not included in Figure 7.

Since it is well known that the bandgap energy (*E_g_*) is a crucial parameter to predict the electronic behavior of semiconductor films, in the next step we obtained the absorption coefficient dependence to estimate the optical bandgap energy of the films obtained at 400, 500, and 600 °C using Tauc’s equation for direct allowed bandgap (1) [60,61]
(*α hν*)^2^ = *B*(*hν* − *E_g_*)(1)
where *α* is the absorption coefficient, *hν* is the incident photon energy, and *B* is a constant. The optical band gap was then estimated by plotting (*αhν*)^2^ against the photon energy *(hν)* and extrapolating the linear portion of the curve to the photon energy axis. Tauc’s plots of films (**1a–1c** and **1e**) are presented in Appendix A. The energy bandgap (*E_g_*) value presented in Table 3 is obtained from 3 independent spectra for each film at each annealing temperature, and assuming uniform reflection (around 20%), as usually measured for this kind of polycrystalline films. For illustrative purposes, Tauc’s plots of the films annealed at 600 °C and the graphical summary of the obtained bandgaps are presented in Figure 8.

All the aliphatic films present rather similar behaviors, progressing in a relatively reduced bandgap interval (3.17–3.28 eV) and reaching values between 3.22–3.26 eV when the annealing temperature increases to 600 °C. The estimated standard deviation, σ, is ±0.01 eV, and the differences in the *E_g_* values for each sample do not clearly exceed 3σ. Thus, correlation between *E_g_* and temperature for films **1a–1c** is not possible. These values are smaller than *E_g_* for pure ZnO reported (around 3.37 eV [1,2,3,4]). According to the literature [30], this could be related to the presence of contaminants that introduce shadow p-doping levels into de bandgap [62] or intrinsic defects, such as Zn or O vacancies or interstitials [30]. Films **1e** have *E_g_* clearly smaller than those found for **1a–1c** under identical conditions. This could be ascribed to different causes such as the higher content of C, H, and N impurities of films **1e** compared with **1a–1c** and the presence of the cubic crystalline form. It is well known that bandgap energy for the sphalerite is smaller (3.18 eV) than the one for wurtzite (3.29 eV) phase.

Another useful information that can be extracted from these spectra is related to the structural disorder in thin films. In particular, the tail of the absorption edge indicates the presence of localized states in the bandgap as given by (2) [63,64],
(2)$$\alpha \left(\lambda \right)=\alpha_{0}\exp\left(\frac{h\nu }{E_{U}}\right)$$
where *α*_0_ is a constant and *E_U_* is Urbach’s energy, which is the width of the tail of the localized states corresponding to the optical transition between localized states adjacent to the valence band and extended state in the conduction band which is lying above the mobility edge [64]. Although the poor thickness uniformity observed in all the obtained films lowers the precision in its determination, the values of the Urbach’s energies have been estimated as the reciprocal of the slopes of the linear portion in the lower photon energy region of *ln(α)* vs. *hν* (Table 3). For aliphatic-derived films, their values are between 0.65 (films **1b** annealed at low temperature) and 0.22 eV (films **1a** annealed at high temperature). *E_U_* shows a decreasing trend with increasing T, suggesting less localized states at higher annealing temperatures. The same trend is observed for the aromatic-derived films (**1e**) but with considerably higher *E_U_* values, only approaching to the previous range after annealing at the highest temperature (600 °C).

#### 3.2.2. Photoluminescence (PL)

The photoluminescence spectra obtained for films annealed at 600 °C are presented in Figure 9 (except for sample **1d**, owing to its low quality). For comparison purposes, spectra corresponding to the samples annealed at 300 °C are also included.

For the films annealed at 600 °C, two emission bands can be distinguished, one band in the range 370–420 nm and the other band, with higher intensity, appearing at lower energies extending roughly from 550 to 750 nm. The PL spectra of samples annealed at 300 °C are noisier and clearly different from those at 600 °C, probably owing to the low content of ZnO in the films at this temperature, in good agreement with the literature [65].

The near-band edge (NBE) for UV emission is present in all samples in the low wavelength range, regardless of the annealing temperature. The energy corresponding to this peak can be found in the range 2.95–3.35 eV and seems to increase with the annealing temperature, which matches well with the bandgap energies obtained from UV-VIS spectroscopy. This emission is attributed to excitonic recombination, a process where an electron in the conduction band is recombined with a hole in the valence band. Due to its crystalline structure, wurtzite has three exciton states, at 362, 366, and 367 nm (3.42, 3.39 and 3.38 eV, respectively) [30,31]. The UV emission of our samples at 600 °C (Figure 9 and Appendix A) is slightly red shifted (*ca*. 18 nm), indicating a bandgap narrowing probably due to a high defect level and sub-stoichiometry of materials. Besides, we can see that film **1e** has the lowest NBE PL intensity at 600 °C, probably because it retains more C and N than films **1a–1c** obtained at identical temperature. Furthermore, we can conclude that the increase in annealing temperature enhances the visible photoluminescence.

The broadening of the band in the visible region might be due to the overlap of several emission peaks close to one another. These deep-level emission bands (DLE) are usually attributed to defects of the ZnO material, such as zinc or oxygen vacancies, interstitials, or anti-sites, among others. Although there is no clear consensus on their origin yet, these emissions are reported to depend on the synthesis procedure, morphology, impurities, vacancies, and surface defects. Typically, green–blue luminescence has been attributed to recombination of simple ionized oxygen vacancies [9], very commonly observed in oxygen-deficient ZnO. It is also thought that oxygen-interstitial related defects are responsible for orange–red luminescence. Moreover, some studies have proved that the addition of carbon increases the oxygen vacancies in ZnO by the reduction reaction [10]. Emissions at 400 ≤ λ ≤ 500 nm are usually associated to defects involving zinc [66]. In the PL spectra of our films annealed at 600 °C (Figure 9, left), the main emission occurs at higher wavelengths, suggesting low relevance, if any, of this sort of defect. Therefore, the broad DLE bands observed should be attributed to structural defects of different typology which are relevant in view of their potential interest in specific areas, e.g., as gas sensors [67]. For instance, the presence of neutral, single-, and double-charge oxygen vacancies or interstitials (V_O_, V_O_^+^, V_O_^++^, and O_i_) produce peaks in the ranges 520–570, 570–620, 620–670, and 670–780 nm, respectively, all of them within the 550–750 nm region, where the broad DLE emission bands of our films annealed at 600 °C were observed. These kinds of structural defects are reported to be strongly dependent on the calcination temperature. Since our films were obtained under identical experimental conditions and calcinated at 600 °C, they are well suited to elucidate whether the AA used in the synthesis of the ink could also induce significant variations in the number and type of these defects in the produced film.

In order to clarify the main origin of the DLE bands visible in Figure 9 (left), they were deconvoluted. The calculated wavelet positions and relative contributions of each component are summarized in Table 4. The results obtained show that one (for **1a** and **1c**), two (for **1e**), or even three (for **1b**) components have low contributions (<5%), and, therefore, won’t be included in the following discussion. Comparison of data shows that transitions from the conduction band to O_i_ levels are more favored in the films prepared using an aromatic AA (**1e**, with a total weight of ca. 75%) than in the aliphatic-derived films **1a–1c**, for which the relative weight was clearly smaller (*ca*. 37% for **1a** and below 50% for **1b** and **1c**). This finding suggests that in addition to the changes induced by the use of the aromatic AA **e** on the properties of films **1e** described in the previous sections, it also induces the preferential formation of O_i_ defects in the ZnO film; while for the aliphatic-based films **1a** and **1c**, transitions from the conduction band to deep-level oxygen vacancies (V_O_, V_O_^+^, and V_O_^++^) are predominant (total contributions varying from 61% for **1a** to 48% for **1c**).

## 4. Conclusions

In this work we have prepared and compared the main properties of ZnO-thin films (**1a**–**1e**) obtained by the sol-gel methodology, using ZAD as the Zn(II) source, the aminoalcohols (**a–e**) as stabilizers, and 2-methoxyethanol as solvent, identical molar ratios, reaction period, and deposition method, but different annealing temperatures (T = 300, 400, 500, and 600 °C). The results summarized here provide conclusive evidence of the effect produced by the nature of the aminoalcohols (aliphatic (**a–c**) or the phenyl-containing analogues (**d** and **e**)) in the ZnO formation process as well as on the physical appearance, stability, purity, morphology, and properties of the films (**1a**–**1e**) at each of the selected temperatures.

In general terms, films (**1a–1c**) obtained using the aliphatic AAs exhibit a similar behavior, stability, morphology, composition, and properties, but some of them are markedly different from those of their aromatic partners **1d** and **1e**, processed identically. For instance, films (**1a–1c**) are clearly more stable than **1e** and, especially than **1d**, which degrades and crashes quickly even at low annealing temperatures, limiting its potential interest. Moreover, the cubic and hexagonal forms of ZnO coexist, in different molar rations (depending on the temperature), in the films prepared from the aromatic aminoalcohols (**d** and **e**); while evidence of the presence of the cubic form was not detected in any of the aliphatic films **1a–1c**. This finding suggests that the aromatic AAs induce the preferential formation of the sphalerite form at low annealing temperatures, that transforms gradually into the wurtzite ZnO phase as the calcination temperature increases. Moreover, the energy band gap (*E_g_*) and the Urbach’s energy (*E_U_*) of the aliphatic-based films are higher and smaller, respectively, than the value obtained for **1e** under identical conditions. Our films exhibit intense emission bands in the visible region, but the analyses of spectral data suggest that the main contribution to the emission spectra is due to transitions from the conduction band to the oxygen-interstitial levels (for **1e**) or to the deep level-oxygen vacancies for **1a–1c**. This indicates that the characteristics and properties of the AA are also important as to modify the type of structural defects and the main origin of the emissive properties of the ZnO films. It should be noted that the results obtained for film **1e** at 600 °C open up a new path to explore other aminoalcohols closely related to aminobenzyl alcohol. The incorporation of substituents with different electron donor/electron acceptor properties on different sites of the phenyl ring may allow to control or tune the bandgap energy and other relevant parameters, as the type and predominance of structural defects present in the final ZnO films, that are important in view of their potential utility to achieve new technological devices.

## Figures and Tables

**Figure 1 nanomaterials-13-01057-f001:**
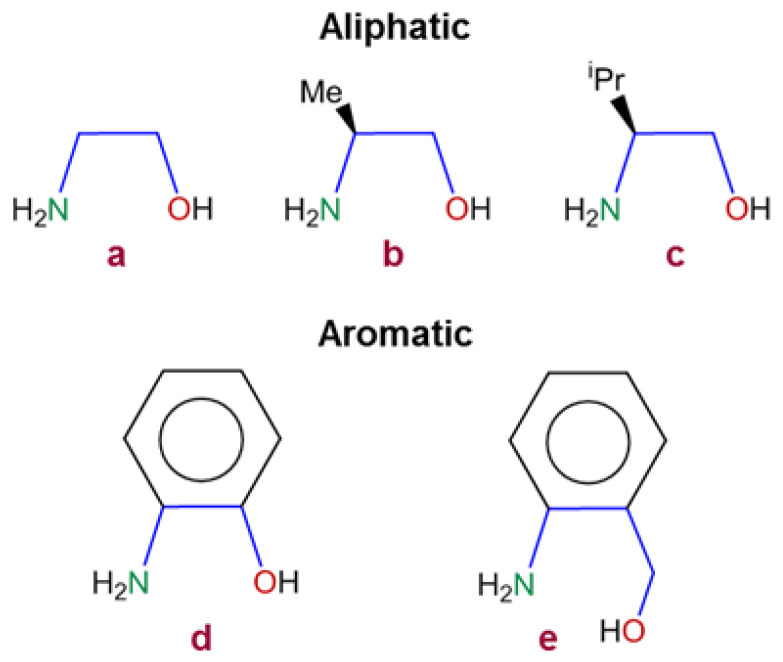
Molecular structure of the AAs **a**–**e** used as precursors to generate the ZnO thin films **1a**–**1e**.

**Figure 2 nanomaterials-13-01057-f002:**
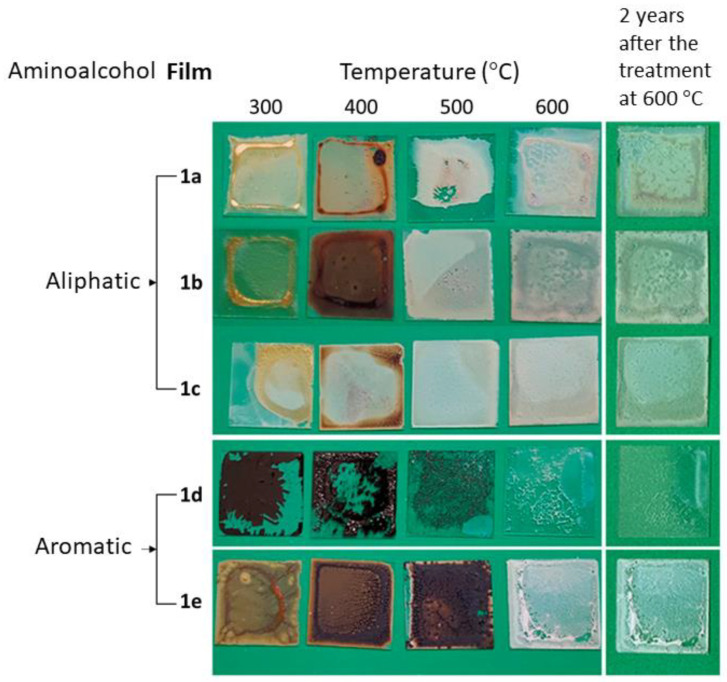
Physical appearance of the films **1a**–**1e** annealed at 300, 400, 500, and 600 °C, and after two years’ storage at room conditions in the dark.

**Figure 3 nanomaterials-13-01057-f003:**
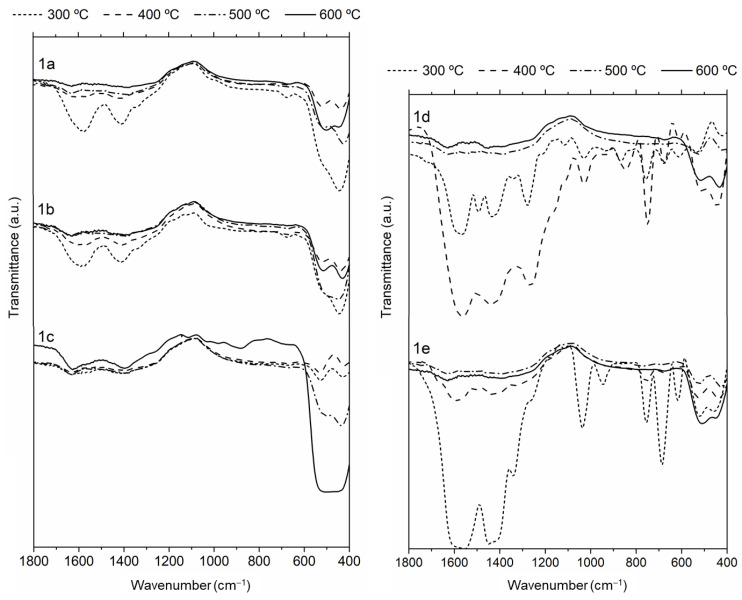
IR spectra of films **1a**–**1e** at different annealing temperatures.

**Figure 4 nanomaterials-13-01057-f004:**
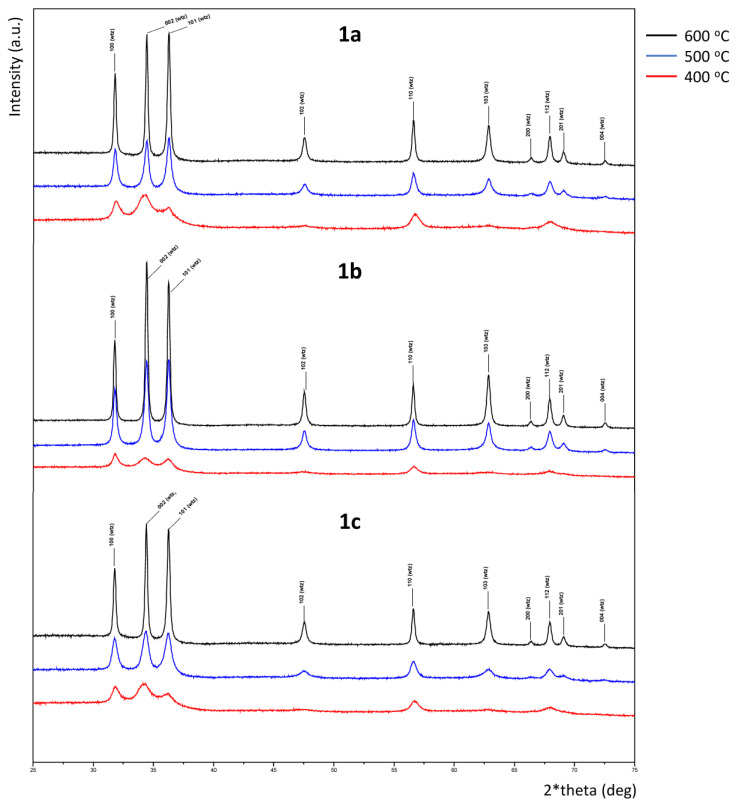
Indexed XRD patterns, in the main angular range from 25 to 75 deg 2θ, of the aliphatic ZnO films **1a–1c** (up) and the aromatic ones **1d–1e** (down) obtained at 400, 500, and 600 °C.

**Figure 5 nanomaterials-13-01057-f005:**
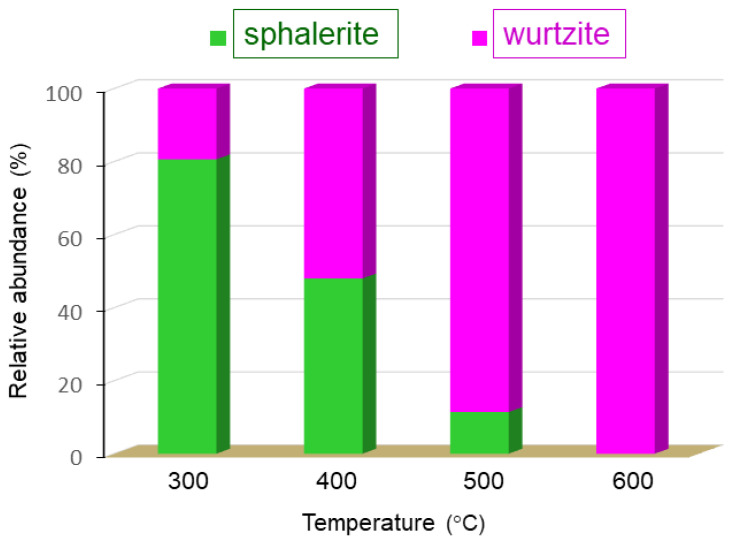
Variation of the relative abundance of the sphalerite and wurtzite forms of ZnO present in film **1e** at each annealing temperature.

**Figure 6 nanomaterials-13-01057-f006:**
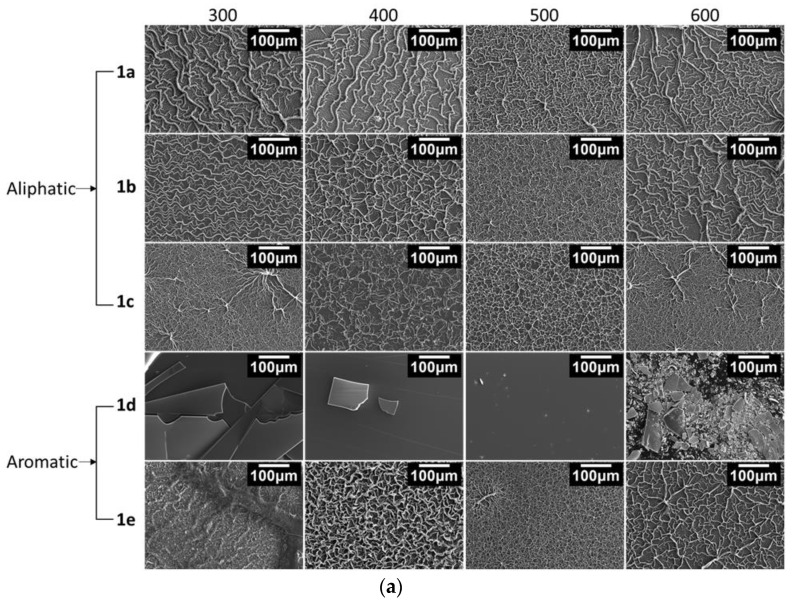
(**a**) Secondary-electron (topology) SEM images of the films. (**b**) Backscattered-electron (composition) SEM images of the films. (**c**) Secondary-electron (topology) SEM images of the films (higher magnification). The insert corresponds to the cross-section of film **1b** annealed at 600 °C.

**Figure 7 nanomaterials-13-01057-f007:**
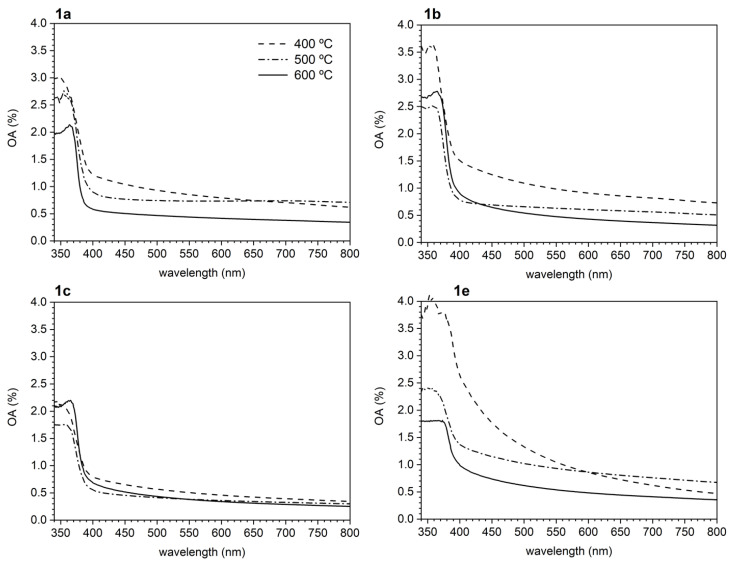
UV-VIS absorption spectra of films **1a–1c** and **1e** at every annealing temperature.

**Figure 8 nanomaterials-13-01057-f008:**
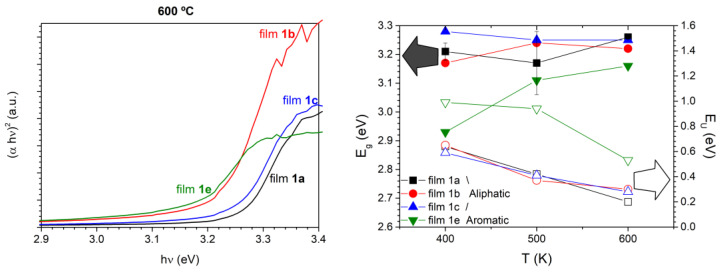
Tauc’s plots of the films annealed at 600 °C (**left**) and graphical summary of gap and Urbach’s energies for every sample annealed at 400, 500, and 600 °C (**right**). Error bars correspond to data dispersion due to different sets of measurements, and for clarity reasons are only included for **1a** *E_g_*.

**Figure 9 nanomaterials-13-01057-f009:**
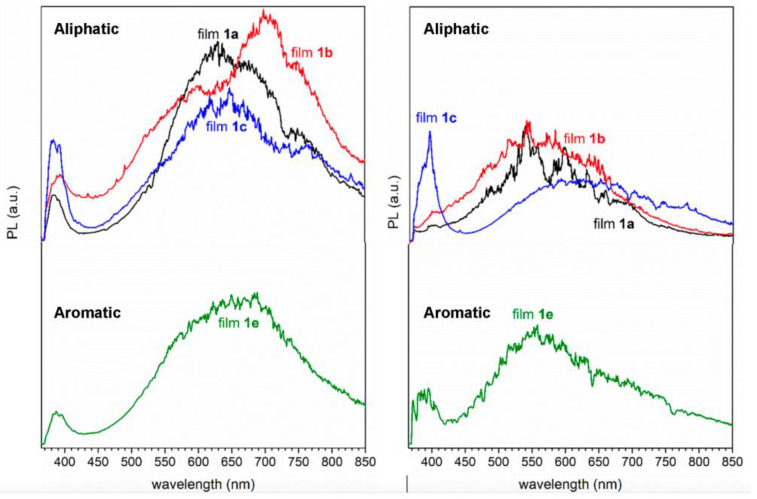
Photoluminescence spectra of the films annealed at 600 °C (**left**) and at 300 °C (**right**), recorded at room temperature.

**Table 1 nanomaterials-13-01057-t001:** Summary of the results obtained from the elemental analyses of **1a**–**1e** films after heating at different temperatures (T) (the standard deviation parameter of data is ±0.2%).

Aminoalcohol	Film	Elemental Analyses (%)	Temperature (°C)			
			300	400	500	600
Aliphatic	**1a**	N	1.23	1.34	0.06	0.01
		C	6.90	6.89	2.94	0.24
		H	0.91	0.53	0.53	0.00
	**1b**	N	1.32	1.01	0.04	0.01
		C	11.96	5.83	0.80	0.27
		H	1.28	0.72	0.13	0.00
	**1c**	N	0.68	0.58	0.09	0.01
		C	12.28	5.80	3.64	0.41
		H	1.34	0.00	0.73	0.00
Aromatic	**1d**	N	5.18	4.49	3.03	0.01
		C	42.23	33.27	20.70	0.71
		H	3.21	2.35	1.88	0.00
	**1e**	N	2.24	4.43	0.57	0.43
		C	32.16	29.32	7.14	3.49
		H	3.54	2.03	0.86	0.00

**Table 2 nanomaterials-13-01057-t002:** Summary of the results obtained from Rietveld XRD studies of films **1a**–**1e** after annealing at 400, 500, and 600 °C: lattice parameters (*a* and *c* (in Å)) and *c/a* ratio, ZnO phase present (% in weight), average crystal size (*D* in nm), and lattice strain (*ε* ×10^–3^).

Aminoalcohol		T	*a*	*c*	*c/a*	Phase, %	*D*	*ε* × 10^−3^
Aliphatic	**1a**	400	3.2398	5.2410	1.618	wurtzite, 100	12.6	2.8
		500	3.2485	5.2096	1.604		14.4	0.3
		600	3.2494	5.2079	1.603		34.4	0.6
	**1b**	400	3.2453	5.2319	1.612	wurtzite, 100	12.7	2.2
		500	3.2487	5.2088	1.603		16.7	0.3
		600	3.2495	5.2065	1.602		37.6	0.6
	**1c**	400	3.2418	5.2423	1.617	wurtzite, 100	12.8	2.7
		500	3.2467	5.2125	1.605		10.3	0.4
		600	3.2490	5.2075	1.603		32.0	0.6
Aromatic	**1d**	400	3.2526 4.5656	5.2105 -	1.602 -	wurtzite, 77.5 sphalerite, 22.5	26.1 10.6	0.1 -
		500	3.2493	5.2083	1.603	wurtzite, 100.0	10.5	0.0
		600	3.2490	5.2066	1.603	wurtzite, 100.0	45.8	0.3
	**1e**	400	3.2494 4.5687	5.2103 -	1.603 -	wurtzite, 52.2 sphalerite, 47.8	25.3 8.3	0.3 -
		500	3.2488 4.5632	5.2137 -	1.605 -	wurtzite, 88.6 sphalerite, 11.4	12.1 4.7	0.0 0.0
		600	3.2488	5.2061	1.602	wurtzite, 100.0	30.4	0.5

**Table 3 nanomaterials-13-01057-t003:** Bandgap (*E_g_*) and Urbach’s (*E_U_*) energies (both in eV) of the films obtained at different annealing temperatures.

Aminoalcohol	Film	Temperature (°C)					
		400		500		600	
		*E_g_*	*E_U_*	*E_g_*	*E_U_*	*E_g_*	*E_U_*
Aliphatic	**1a**	3.21	0.64	3.17	0.42	3.26	0.22
	**1b**	3.17	0.65	3.24	0.37	3.22	0.30
	**1c**	3.28	0.59	3.25	0.41	3.25	0.28
Aromatic	**1e**	2.93	0.99	3.11	0.94	3.16	0.53

**Table 4 nanomaterials-13-01057-t004:** Wavelet position, full width at half maximum (FWHM), area, relative weight, and possible assignment of each DLE component (according to [63]) obtained after Gaussian deconvolution for **1a–1c** and **1e** films annealed at 600 °C.

Aminoalcohol	Film	Photoluminescent Band				Origin
		Center (nm)	FWHM (nm)	Area (a.u.)	%	
Aliphatic	**1a**	570.0	51.2	6.94	10.5	V_O_ or V_O_^+^
		627.3	84.2	33.92	51.0	V_O_^++^
		687.6	59.6	10.92	16.4	O_i_
		714.4	17.1	1.07	1.6	O_i_
		750.3	83.2	13.61	20.5	O_i_
	**1b**	534.5	46.6	1.27	2.2	V_O_
		601.5	115.9	24.97	43.6	V_O_^+^
		680.3	57.5	1.26	2.2	O_i_
		710.5	28.5	2.78	4.8	O_i_
		742.1	99.2	27.04	47.2	O_i_
	**1c**	593.3	56.8	2.78	4.9	V_O_^+^
		593.3	125.7	12.45	22.2	V_O_^+^
		658.2	89.97	14.63	26.0	V_O_^++^
		724.9	277.6	21.58	38.4	O_i_
		763.4	69.4	4.76	8.5	O_i_
Aromatic	**1e**	574.0	89.1	14.67	20.7	V_O_^+^
		596.9	17.9	0.32	0.4	V_O_^+^
		628.3	41.8	2.55	3.6	V_O_^++^
		676.3	83.2	11.19	15.8	O_i_
		696.6	200.7	42.19	59.5	O_i_

## Data Availability

Not applicable.

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
