# Peer review of "Influence of the Nature of Aminoalcohol on ZnO Films Formed by Sol-Gel Methods"

_nanomaterials, 2023, doi:10.3390/nano13061057_

Round 1
Reviewer 1 Report
I have reviewed the submitted work to Nanomaterials/MDPI entitled “Influence of the nature of the amino alcohol on ZnO films formed by sol-gel methods” by Vila et al. Sol-gel derived ZnO is one of the most attractive oxides due to its excellent physicochemical properties including UV absorption behavior. Sol-gel synthesis for producing high-quality structures has gained wide attraction among researchers. The authors have obtained this by using zinc acetate dihydrate, methoxy ethanol as salt, and ethanolamine as a stabilizer. The presence of amines increases the alkalinity of the medium. Control of the pH of the solution by adding amino alcohols allows for a particular prediction of the layer’s behavior. The influence of aliphatic versus aromatic on the ink stability is discussed on various parameters. The manuscript itself is written OK but the main concern of this paper and the areas that require clarification are given below.
· In the submitted work, it appears there is no standalone novelty rather some incremental work has been carried out. Strictly speaking, the manuscript should be rejected. Unless otherwise, the authors completely revise the introduction and make substantial efforts to reinforce the objectives of this work and how it differs from the work already published. (including doi.org/10.3390/gels8080512; and doi.org/10.1021/acs.jpcc.7b09935).
· Line 185, how the annealing temperature increased the content of N, C, and H? in contrast to line 193?
· In section 3.1.2, the X-ray diffractogram of ZnO peaks and its corresponding stability structure can be referred back to the relevant literature reported for Zinc Oxide such as DOI 10.1149/1.3387672; doi.org/10.1016/j.ijhydene.2010.04.143. Is there a formation of Zn(OH)2 peaks?
· All the SEM images need to be provided with a scale bar.
· Does the size of the crystallites have a direct relation to the concentration or the type of stabilizer or its concentration?
· The fractal structure and homogeneity of the crystallites can be detailed in the SEM-related section.
· What is the effect of annealing temperature on observed optical properties?
· A higher concentration of amino alcohols leads to higher Eg values and lower transmittance. Any threshold for it?
Author Response
Thank you very much for your comments concerning our manuscript entitled
“Influence of the nature of the aminoalcohol on ZnO films formed by sol-gel methods”,
coded nanomaterials-2230109.
We have read carefully your report and modified the manuscript accordingly. In a separate document, we include a detailed description of our answers to your comments and suggestions, the actions taken, as well as the location where you can find the changes in the revised version, where they have been highlighted by the Track Changes activation by Balloons in Review tag in Word.
We sincerely appreciate your suggestions, which definitvely improve the quality and clarity of our manuscript, and hope that this new version could be accepted for publication in Nanomaterials.

Reviewer 2 Report
The manuscript “Influence of the nature of the aminoalcohol on ZnO films 2 formed by sol-gel methods” by Anna Vilà et.al. is suitable for publication in Nanomaterials after minor revision
Incorrect link to the Supplementary data
At www.mdpi.com/article/10.3390/nano12091508/s1 one will find
Supplementary Materials Preparation, Characterization and Application of Epitaxial Grown BiOBr (110) Film on ZnFe2O4 Surface with Enhanced Photocatalytic Fenton Oxidation Properties.
Page 4, row 126 “… a JEOL J-6510 …”
Reviewer’s comment: should be JEOL JSM-6510
Author Response

(The authors gave the same response as above.)

Reviewer 3 Report
This manuscript presents the influence of the nature of the aminoalcohol on ZnO films 2 formed by sol-gel methods. There are a few remarks that, I hope, can help the authors to improve the text:
1. What was the thickness of the samples?
2. For me, sample 1a at 600 degree and sample 1a after 2 years at 600 degree are not the same.
3. Why is there no relationship between Eg and temperature for samples 1a and 1b (see Table 3)?
4. At what temperature the photoluminescence measurements were made?
5. Authors should read and cite the following article: Opto-Electron. Rev. 28, 182-190 (2020) and Materials 13, 2559 (2020).
8. More interpretation of the results should be added.
Summarize, this review article is interesting, and it could be published in Nanomaterials but after major revision.
Author Response

(The authors gave the same response as above.)

Reviewer 4 Report
The paper presents comparative studies for the ZnO sol-gel films prepared using two different sol compositions. The resulted films where characterised for the structural and optical properties and the differences are discussed. I have several observations and suggestions.
I think that the XRD spectra must be exposed also in the paper. In this way can be visible if some amorphous component is presents. I suppose that after 400C annealing some amorphous component remain in the film. I also suppose that the crystallization process is different on the ridges like in the plateau areas. Probably some TEM studies can reveal better the local nanometric structure after crystallization. I am curious if the authors have an explanation for the nucleation of the cubic phase in the case of aromatic sol and not in the aliphatic sol preparation.
The ZnO films are quite thick and are possible some textures after crystallization and growth of the ZnO crystallites at least for the 600 C annealing. There is some texture visible in the XRD spectra?
Author Response

(The authors gave the same response as above.)

Round 2
Reviewer 1 Report
This reviewer went through the revised version of the submitted article. The authors have addressed the queries satisfactorily. Therefore, the revised version is suitable to publish.
Reviewer 3 Report
Accepted responses to reviewer comments.